# Synergistic Effects of Metformin and Trastuzumab on HER2 Positive Gastroesophageal Adenocarcinoma Cells In Vitro and In Vivo

**DOI:** 10.3390/cancers15194768

**Published:** 2023-09-28

**Authors:** Jin-Soo Kim, Mi Young Kim, Sungyoul Hong

**Affiliations:** 1Department of Internal Medicine, Seoul National University Boramae Medical Center, Seoul 07061, Republic of Korea; moung9805@naver.com; 2College of Pharmacy, Seoul National University, Seoul 08826, Republic of Korea; sungyoul@snu.ac.kr

**Keywords:** gastric cancer, ErbB-2 receptor, trastuzumab, metformin

## Abstract

**Simple Summary:**

Metformin is widely used to treat type 2 diabetes with minimal side effects. Trastuzumab (Tmab) in combination with chemotherapy has been the standard treatment for HER2+ gastric cancer (GC) for the last two decades. Unfortunately, the use of subsequent HER2 targeted agents have not proven to be beneficial for HER2+ GC, although these agents are effective and currently being used for management of HER2+ breast cancer. In the present study, metformin enhanced the efficacy of trastuzumab in HER2+ GC cells. Therefore, metformin may be a safe and effective therapy for treatment of HER2+ gastric cancer, providing additional benefits at a low cost.

**Abstract:**

The incidence of HER2 amplification in advanced gastroesophageal adenocarcinoma (GC) reportedly ranges between 10% and 20%, depending on the population studied and the geographical region. Trastuzumab (Tmab) is the standard treatment for GCs with HER2 amplification. Metformin, a widely used antidiabetic drug, is an activator of AMP kinase that can affect the mTOR signaling pathway. The following GC cells were evaluated: HER2+ NCI-N87, YCC-19, YCC-38, OE19, OE33, and HER2- AGS. The effects of Tmab and metformin on these cell lines were assessed as single agents and in combination using cell viability assays, Western blotting, and xenograft models. Metformin induced phosphorylation of AMP kinase in all tested GC cells and dephosphorylation of mTOR in Tmab-sensitive GC cells. We observed that treatment with Tmab in combination with metformin induced a significant decrease in the number of colonies formed on soft agar by N87, YCC-19, YCC-38, and OE19 cells (88%, 95%, 73%, and 98%, respectively), in comparison to the number formed by control cells or cells in the single-treatment groups. No growth inhibition was detected in OE33 cells treated with Tmab alone. Combination with metformin resulted in decreased phosphorylation of HER2 and its downstream targets, AKT and ERK, in Tmab-sensitive HER2+ cells. Phospho-receptor tyrosine kinase (RTK) arrays were used to profile the phospho-proteome, which demonstrated a synergistic decrease in phosphorylation of EGFR, HER2, and HER3. Furthermore, the combination of Tmab and metformin exhibited enhanced antitumor effects in a xenograft model. Collectively, these data suggest that Tmab and metformin act synergistically in HER2+ GC cells. Since metformin is widely used and relatively non-toxic, its addition to the therapeutic regimen along with Tmab could enhance the clinical efficacy in patients with HER2+ GC.

## 1. Introduction

Gastroesophageal adenocarcinoma (GC) is a heterogeneous disease characterized by distinct molecular, anatomical, and histological subtypes. The human epidermal growth factor receptor-2 (HER2) is an important target for treating patients with advanced GC. The pivotal ToGA (Trastuzumab for Gastric Cancer) trial compared the efficacy of chemotherapy alone with trastuzumab (Tmab) in combination with chemotherapy in patients with HER+ GC [1]. This trial confirmed the benefits of Tmab treatment in patients with HER2+ GC in terms of response rates and survival. In recent trials, patients with HER2+ breast cancer (BC) have been reported as deriving clinically meaningful benefits from subsequent administration of HER2 inhibitors, such as lapatinib, pertuzumab, and T-DM1. These observations have prompted clinical trials of these newer HER2-directed therapies for HER2-positive GC. However, the outcomes of these drugs did not meet the anticipated expectations: lapatinib in the second-line [2] and first-line settings [3]; pertuzumab in the first-line setting [4]; and T-DM1 in the second line setting [5]. Even though HER2-targeted therapies for advanced GC have not been very successful, recent trials with trastuzumab deruxtecan have led to significant improvements in response and overall survival (OS), as compared with standard therapies, among pretreated patients with HER2+ GC [6]. These findings suggest that selecting the patient population to target with HER inhibitors may be critical for achieving clinical benefits. There is an unmet medical need for the systemic treatment of patients with HER2+ GC, since several HER2-targeted agents have failed, even though these agents are widely used for patients with HER2+ BC.

Molecular profiling of human GCs have been performed to improve the therapeutic strategy for this recalcitrant cancer, which has revealed distinct subtypes that are characterized and often driven by altered signaling pathways [7,8,9]. One such signaling pathway that has been shown to be altered in several types of human cancers is the phosphatidylinositol 3-kinase (PI3K)/AKT/mammalian target of rapamycin (mTOR) pathway. In the TCGA gastric cancer dataset, high mTOR activity was observed in the proliferative clusters using reverse phase protein array analysis, and approximately half of HER2-amplified GC showed increased mTOR expression [7]. Although the mTOR inhibitor, Everolimus, as a single agent did not significantly improve the overall survival (OS) of patients with AGC in whom progression occurred after one or two lines of previous systemic chemotherapy [10], inhibition of the mTOR pathway could be a strategy to improve Tmab efficacy in patients with HER2+ GC.

Metformin is an oral biguanide that was introduced into clinical practice in the 1950s for treatment of type 2 diabetes mellitus. However, the exact mechanism of action of metformin is not fully understood. Interestingly, the anticarcinogenic activity of metformin has been attributed to several mechanisms, including activation of liver kinase B1 (LKB1)/AMPK pathway, induction of cell cycle arrest and/or apoptosis, inhibition of protein synthesis, inhibition of unfolded protein response, activation of the immune system, and possible eradication of cancer stem cells [11]. Metformin reportedly inhibits cell proliferation and tumor growth in GC, possibly by suppressing cell cycle-related molecules via alteration of miRNAs [12]. Decrease in proliferation observed in vitro and in vivo has been related to cell cycle arrest in MKN74 cells, which is now known to harbor HER2 amplification. AMPK/mTOR-mediated inhibition of survivin has been suggested to partially contribute to metformin-induced apoptosis in gastric cancer cells [13]. Phenformin, a mitochondrial inhibitor and analog of metformin, selectively induces apoptosis in LKB1-deficient NSCLC cells [14]. However, it was removed from clinical use for type 2 diabetes due to fatalities resulting from lactic acidosis. The effects of metformin on AMPK activation also result in reduced mTOR signaling and protein synthesis [15]. The use of metformin has been associated with improved survival in patients with different types of cancers, including those with GC after gastrectomy [16], those with diabetes who were diagnosed with GC in a Swedish population-based cohort study [17], those with HER2 positive BC in a population study [18], and laboratory studies [19,20,21]. This evidence has led to clinical trials in patients with HER2+ or HER2- BCs [22,23,24,25], which should be interpreted with caution due to the relatively small size of the trials.

This study aimed to examine the effects of combined treatment of metformin with Tmab on HER2+ GC cell lines. Based on previous studies, we focused on the modulation of the mTOR pathway by metformin in HER2+ GC cells. For this purpose, we used a panel of human GC cell lines with a defined spectrum of sensitivity to inhibition of HER2 by Tmab.

## 2. Materials and Methods

### 2.1. Cell Line and Reagents

Among the 37 human gastric and esophageal cell lines (Appendix A), gastric cancer (NCI-N87, YCC-19, YCC-38, and AGS) and esophageal adenocarcinoma (OE19 and OE33) cell lines were selected for this study. AGS and NCI-N87 cell lines were obtained from the Korean Cell Line Bank. YCC-19 and YCC-38 cell lines were provided by Professor Sun-Young Rha (Yonsei Cancer Center, Republic of Korea). YCC-19 and YCC-38 were established at the Yonsei Cancer Center from the malignant ascites of patients with GC. OE19 and OE33 were purchased from Sigma-Aldrich (St. Louis, MI, USA). Detailed information about the purchased cell lines can be retrieved from the vendors. AGS, NCI-N87, OE19, and OE33 cell lines were cultured in Roswell Park Memorial Institute-1640 (RPMI-1640) medium. YCC-19 and YCC-38 cell lines were cultured in Eagle’s Minimum Essential Medium (EMEM) containing 10% fetal bovine serum, 100 units/mL penicillin, and 100 mg/mL streptomycin. Cultured cells were maintained at 37 °C in an atmosphere with 5% carbon dioxide. Tmab and metformin were purchased from Orient Bio (Seongnam-si, Republic of Korea). Passage numbers were recorded and maintained at low levels.

### 2.2. Droplet Digital PCR

The droplet digital PCR (ddPCR) assay was performed by the LOGONE Bio-Convergence Research Foundation (Seoul, Republic of Korea) using the QX200 system (Bio-Rad, Hercules, CA, USA) in accordance with the manufacturer’s recommendations. The reaction mixture in a volume of 20 µL comprised 4× ddPCR multiplex supermix (5 μL) for Probes (Bio-Rad, Hercules, CA, USA) and primer-probe set for the reference genes, which were designed by LOGONE Bio-Convergence Research Foundation (Seoul, Republic of Korea). For HER2 copy number analysis, the concentrations of primers and probes were varied in the range of 900–250 nM. The primer sequences are listed in Appendix A. The entire reaction mixtures with 70 µL of droplet generation oil (Bio-Rad, Hercules, CA, USA) were loaded into the oil well for each channel and placed in the droplet generator. After processing, droplets obtained from each sample were transferred to a 96-well PCR plate. The plates were heat-sealed, placed on a T100 Thermal cycler (Bio-Rad, Hercules, CA, USA), and amplified to the end-point. PCR was performed under the following conditions: Uracil DNA glycosylase reaction at 37 °C for 30 min and DNA polymerase activation at 95 °C for 5 min followed by 40 cycles of PCR amplification (95 °C for 30 s and 58 °C for 60 s), and 4 °C for 5 min, 90 °C for 5 min, 2 °C/s ramp rate at all steps. After PCR, the droplets were counted using the QX200 Droplet Reader. The data obtained were analyzed using QuantaSoft Analysis Pro software 1.7.4.0917 (Bio-Rad, Hercules, CA, USA).

### 2.3. Cell Proliferation Assay

Cell proliferation and viability were assessed using the conventional 3-(4,5-dimethylthiazol-2-yl)-2,5-diphenyltetrazolium bromide (MTT) assay, obtained from Sigma-Aldrich (St. Louis, MI, USA), in accordance with the manufacturer’s protocol. Cells were initially plated in 96-well plates, ranging between 2500 and 10,000 cells per well, and allowed to adhere for a period of 24 h. Subsequently, the cells were exposed to the specified concentrations of Tmab (in µg/mL) or metformin (in mM) within the standard culture medium for a duration of 72 h. The absorbance at 540 nm was measured using a SpectraMax Plus 384 reader (Molecular Devices, San Jose, CA, USA).

### 2.4. Growth in Soft Agar

For anchorage-independent colony formation assay, 2.5 × 10^3^–7.5 × 10^3^ cells were suspended in 0.5 mL of 0.4% top agar that was layered on top of 1 mL of 1% base agar in each well in 12-well multi-well plates, as previously described [26]. The plates were incubated for 10–15 days in a complete medium containing either metformin or Tmab alone, as well as in combination at the specified concentrations. During incubation, the medium was refreshed twice a week, and each agent was added simultaneously. Subsequently, the colonies were stained using crystal violet solution (Cat. No. V5265; Sigma-Aldrich, St. Louis, MI, USA) diluted with distilled water to a concentration of 0.001% and quantified using ImageJ Software 1.53i (National Institutes of Health, Bethesda, MA, USA). To calculate the mean growth inhibition (MGI), we divided the colony count of each treatment group by that of the control group. The anticipated MGI was determined by multiplying the MGIs for individual treatments, and then an index was computed by dividing the expected colony count by the observed colony count. An index value greater than 1 indicated a synergistic effect, whereas an index value below 1 suggested a less-than-additive effect. All assays were performed in triplicates.

### 2.5. Western Blot Analysis

Cells (2–8 × 10^5^) were seeded in 100-mm dishes and exposed to the indicated conditions. Proteins were harvested as previously described [26]. Briefly, total cell lysates were prepared using a modified RIPA lysis buffer containing protease and phosphatase inhibitors. Western blots were performed to compare the expression of relevant signaling pathways, such as the HER2 signaling pathway. The following primary antibodies were purchased from Cell Signaling Technology (Beverley, MA, USA): p-HER2-Tyr 1221/1222 (#2249), HER2 (#2165), p-mTOR-Ser 2448 (#2971), mTOR (#2983), p-EGFR-Tyr 1068 (#2234), EGFR (#2646), p-AMPKα-Thr 172 (#2535), p-AKT-Ser 473 (#4060), AKT (#2217), p-ERK-Thr 202/Tyr 204 (#9106), p-S6-Ser 235/236 (#2211) and S6 (#2217). AMPK (#sc-19128), ERK (#sc-271269), and Actin (#sc-47778) were purchased from Santa Cruz Biotechnology (Dallas, TX, USA), and secondary antibodies were obtained from Thermo Fisher Scientific (Waltham, MA, USA).

Equal amounts of protein (25 μg) were subjected to 6–12% SDS-PAGE and electrically transferred onto nitrocellulose (Cat. No. 66485; Pall Corporation, Port Washington, NY, USA). Membranes were blocked using a blocking buffer (consisting of 5% non-fat dry milk in Tris-buffered saline containing 0.01% Tween-20, TBST) for 1 h at room temperature. Subsequently, the membranes were subjected to overnight incubation with primary antibodies at 4 °C, which were appropriately diluted in 5% BSA in TBST at a ratio of 1:1000. Following primary antibody incubation, the membranes were washed multiple times with TBST and exposed to the corresponding horseradish peroxidase-conjugated secondary antibodies, which were diluted in 3% nonfat dry milk in TBST at a ratio of 1:5000, for 1 h at room temperature. After multiple washes with PBST, the membranes were visualized using an enhanced chemiluminescence detection kit (Amersham Biosciences, Arlington Heights, IL, USA). A Western blot of β-actin was included as a loading control. PageRuler™ Plus Prestained Protein Ladder (Cat. No. 26616; Thermo Fisher Scientific, Waltham, MA, USA) was used. The chemiluminescence was recorded using the ChemiDoc™ Imaging System (Cat. No. 12003153; Bio-Rad, Hercules, CA, USA).

### 2.6. Proteome Profiling with Phospho-RTK Array

We analyzed the expression and activation of receptor tyrosine kinases (RTKs) as well as their downstream signaling pathways using the Proteome Profiler Human Phospho-RTK Array Kit from R&D Systems (Minneapolis, MN, USA). Cell lysates (500 μg) were incubated overnight with the provided membrane, in accordance with the manufacturer’s instructions. To quantify the activation of RTKs, we calculated the densitometric ratio of duplicate spots compared to the loading controls on the RTK array using ImageJ software 1.53i. The intensity values of the probes and local background were subjected to a log2 transformation to achieve a more symmetrical distribution. Subsequently, the fold change was calculated by determining the difference between the transformed values.

### 2.7. Xenograft Mouse Model

All animal procedures were performed in accordance with the protocols approved by the Seoul National University Institutional Animal Care and Use Committee (IACUC). The animal experiments were approved by the IACUC of Seoul National University (SNU-171123-3-1). Five-week-old female athymic nude mice were obtained from Orient Bio (Seongnam-si, Republic of Korea) and acclimated for 14 days. Among the GC cell lines, we used NCI-N87 cells, which have been previously tested in many xenograft experiments. Human NCI-N87 gastric cancer cells were cultured in RPMI-1640 (10% FBS) medium and resuspended in 50% Cultrex Basement Membrane Extract in phosphate buffered saline. Viable NCI-N87 cells were subcutaneously injected into the right flank of the mice at a concentration of 2 × 10^6^ cells per mouse. When the average tumor size reached approximately 150 mm^3^, the mice were randomly divided into vehicle control or treatment groups, with six mice belonging to each group. The day of randomization was defined as Day 0. Tmab at 10 mg/kg, as previously reported [27], was administered intraperitoneally every three days (Q3D) to mice for three weeks. Formulated metformin at 200 mg/kg, as previously reported [28], was administrated intraperitoneally every day (5D+/2D−) to the mice for three weeks. The size of the tumor was measured twice weekly using digital calipers, and the volume of the tumor was calculated using the following formula: volume (mm^3^) = length (mm) × width (mm)^2^ × 0.5. All mice were observed daily during the treatment period and euthanized in accordance with the guidelines provided by the IACUC of the SNU.

### 2.8. Statistical Analysis

Analyses were conducted using SPSS version 27.0 (IBM SPSS Statistics for Windows, Version 27.0. Armonk, NY, USA: IBM Corp.). Data were presented as means ± standard error (SE). The statistical significance of differences was determined using the Kruskal–Wallis test, One-way Analysis of Variance (ANOVA), and Student’s *t*-test, as appropriate. All statistical analyses were two-tailed. A *p* value of less than 0.05 was considered statistically significant. The combined drug effects were analyzed by calculating the combination index (CI) [29] using CalcuSyn software 2.11 (Biosoft, Cambridge, UK). CI values of <1, 1, and >1 indicated synergistic, additive, and antagonistic effects, respectively.

## 3. Results

### 3.1. Identification of HER2 Positivity and Determination of Sensitivity to Tmab

Among the 37 GC cell lines, we confirmed HER2 overexpression and/or amplification, as there have been conflicting reports on HER2 positivity and sensitivity to Tmab in each GC cell line. To characterize the gastric cancer cells, we performed immunohistochemistry for HER2 expression on cell blocks. Moreover, ddPCR was performed to determine HER2 gene copy numbers. Among these, 10 GC cell lines showed increased HER2 copy numbers using ddPCR (Appendix A). AGS gastric cells are human gastric adenocarcinoma-derived cell lines, and these were selected as the HER2-negative model. The following GC cell lines were evaluated for subsequent experiments: HER2+ NCI-N87, YCC-19, YCC-38, OE19, OE33, and HER2- AGS. As determined by the MTT and anchorage-independent colony-forming assays, GC cells were responsive to Tmab or metformin with variable sensitivity (Figure 1A,B).

When we tested Tmab on the selected GC cell lines using the MTT assay, the majority of the HER2+ GC cells did not reach IC_50_, even at 10 μg/mL (Appendix A). In contrast, YCC-38 cells were relatively resistant to Tmab (Figure 1B) in the soft agar colony formation assay, whereas NCI-N87 and YCC-19 cells were relatively sensitive to Tmab. OE33 cells have been shown to have reduced sensitivity to HER2 inhibitor, Lapatinib, when used as a single agent [30,31]. Our data also confirmed the relative resistance to Tmab using the MTT assay (Figure 1A). Unfortunately, the OE-33 cells did not form assessable colonies on soft agar, and we could not examine these cells in the colony-forming assay. HER2- AGS cells did not respond to Tmab, but showed moderate sensitivity to metformin. In addition, metformin induced a dose-dependent inhibition of HER2+ GC cell growth. Interestingly, we observed a dose-dependent increase in phosphorylated AMPK (pAMPK) and a dose-dependent decrease in phosphorylated HER2 (pHER2) in HER2+ GC cells (Figure 1C). These data suggest that metformin treatment can inhibit HER2+ GC cell growth when used as a monotherapy.

### 3.2. Metformin Induced Phosphorylation of AMP Kinase and Dephosphorylation of mTOR in Tmab Sensitive GC Cells

To evaluate the effect of metformin on HER2-positive cancer cells, signals related to HER2 expression were assessed using Western blotting. The GC cells were treated with 10 mM metformin for 24 h. Since the anti-cancer mechanisms of metformin involve direct and indirect AMPK-dependent and -independent inhibition of mammalian target of rapamycin (mTOR) [11,32], we focused on the changes in mTOR and downstream ERK or S6 phosphorylation. In HER2+ GC cells, metformin effectively downregulated the phosphorylation of HER2, EGFR, and mTOR. Even in HER2- AGS cells, metformin induced pAMPK, but downstream the mTOR phosphorylation was not strongly affected (Figure 2). Interestingly, Tmab-resistant OE33 cells also did not show decreased mTOR phosphorylation.

### 3.3. Effect of Metformin in Combination with Tmab on GC Proliferation and Anchorage Independent Colony Forming Ability

We then assessed the growth-inhibitory effects of combinations of metformin with Tmab. GC cells were treated with Tmab (0.01, 0.1, and 1 μg/mL), metformin (0.1, 1, and 10 mM), or the combination. As shown in Figure 3A, 72-h exposure to metformin and Tmab resulted in a clear synergism in NCI-N87, OE19, and YCC-38 cells, with CI less than 0.5. For YCC-19, OE33, and AGS cells, the CI was greater than 1, suggesting an antagonistic effect. We also confirmed the significant synergism of the combination of these two agents using soft agar colony-forming assays (Figure 3B).

These drugs exhibited synergistic inhibitory effects on colony formation on soft agar (Table 1).

### 3.4. Tmab and Metformin Are Synergistic in Tmab-Sensitive HER2+ GC Cells

As shown in Figure 1C, metformin induced robust activation of AMP kinase, which resulted in suppression of ERK and S6 signaling. Importantly, the combination of Tmab and metformin synergistically dephosphorylated HER2, resulting in strong inhibition of ERK and S6 signaling (Figure 4A). In Tmab-resistant OE33 cells, the combination effectively inhibited ERK and S6 signaling, but the decrease was modest in comparison to that of other HER2+ GC cells. Metformin also suppressed EGFR phosphorylation. Phospho-RTK array was used to assess the phosphorylated levels of various RTKs in N87, OE19, YCC-19, and OE33 cell lysates treated with Tmab, metformin, or their combination for 24 h before harvest. Consistent with previous results, we observed strong HER2 activity in HER2+ GC cells at baseline, and the combination of Tmab and metformin effectively suppressed phospho-HER2, which was demonstrated using a commercially available phospho-RTK array kit (Figure 4B,C).

Collectively, these findings suggest that the combination of Tmab and metformin exhibited clear synergism in HER2+ GC cells in vitro.

### 3.5. Combination of Tmab and Metformin Shows Superior Efficacy in Suppression of Tumor Growth In Vivo

Next, we tested our hypothesis that a combination of Tmab and metformin would be effective in vivo. NCI-N87 xenografts were treated with Tmab and/or metformin for 21 days. NCI-N87 tumor growth was modestly reduced by Tmab or metformin alone. The combination of these two agents resulted in significantly better inhibition of the xenograft (Figure 5A), while the body weights of the mice were similar among the allocated groups (Figure 5B). From D24 to D31, the sizes of the xenografts among the four groups differed significantly (D24, *p* = 0.02; D28, *p* = 0.023; D31, *p* = 0.03; Kruskal–Wallis Test).

With the combination treatment, the tumor measurement showed an approximately 43% reduction compared to the control at D24. These findings suggest that the combination of Tmab and metformin shows clear synergism in HER2+ GC cells in vivo.

## 4. Discussion

We showed that metformin could synergize with Tmab for treatment of HER2+ GC using in vitro and in vivo experiments. The xenograft experiment showed that the combination treatment resulted in an approximately 43% decrease in comparison to the control group. Metformin induced AMPK phosphorylation and HER2 dephosphorylation in HER2+ GC cells. A xenograft study using NCI-N87 cells confirmed the synergistic interaction of metformin and Tmab without significant toxicity.

Except for the recent approval of trastuzumab deruxtecan in pre-treated patients with HER2 + GC [6], other HER2-targeted agents have not been proven to be effective in patients with HER2+ GC during the last decade, even though these agents are effective and currently used in patients with HER2+ BC. The reason why BC and GC respond differently to these agents is beyond the scope of this study. However, we would like to stress that there is a strong unmet need for effective therapeutic interventions for patients with HER2+ GC. The differential efficacy of anti-HER2 agents has been partly discussed in recently published review articles [33,34]. To improve the efficacy of Tmab for patients with HER2+ GC, one straightforward strategy is the addition of other agents, as Tmab remains the main armamentarium for this condition. Furthermore, we should utilize well-characterized HER2+ GC models, as HER2+ GC has shown differential sensitivities to HER2 targeting agents in clinical trials.

Our data showed that metformin could decrease the phosphorylation of HER2 in HER2+ GC cells. This could be the mechanism of synergy with Tmab in this model. Although several preclinical suggestions for this combination have been presented for HER2+ BC [32,35], no preclinical research has been conducted on HER2+ GC using this combination. Kato et al. were the first to study the effects of metformin in vitro and in vivo in different GC cell lines (MKN1, MKN45, and MKN74) [12]. The decrease in proliferation observed in vitro and in vivo was related to cell cycle arrest in MKN74 cells, which are known to harbor HER2 amplification. The authors did not focus on HER2 positivity in this cell line. However, they also showed that metformin reduced the expression of p-EGFR and p-IGF-1R in the protein array, which was in concordance with the present data. Furthermore, metformin has been reported to inhibit the proliferation and growth of human esophageal adenocarcinoma cells [24]. Similar to the results of the present study, the authors showed that metformin inhibited the proliferation of HER2+ esophageal adenocarcinoma OE19 cells in vitro and in vivo. Human phospho-RTK array confirmed that metformin also reduced the levels of expression of p-EGFR, p-HER2, and p-IGF-1R in vitro. More data were derived from the HER2+ BC studies. In the HER2+ BC model, 4T1 tumor tissue from metformin-treated mice showed lower levels of phospho-HER2 [36]. In addition, metformin significantly decreased the phosphorylation of HER2 protein in vitro in a time-dependent manner. The authors also showed that metformin treatment significantly inhibits 4T1 tumor growth in vivo. Another group suggested that metformin might overcome tamoxifen resistance by inhibiting the expression and signaling of receptor tyrosine kinases HER2 and HER3 in tamoxifen-resistant BC cells [37]. Treatment with metformin induces AMPK activation and reversal of Lapatinib resistance in HER2+ BC MCF-7 cells [38]. Activated AMPK negatively regulates HER2 and EGFR signaling by phosphorylating both the proteins at regulatory sites in a BC model [39]. These data suggest that metformin enhances the efficacy of Tmab by inhibiting HER2 signaling in HER2+ cancer models. However, the exact mechanism of HER2 modulation by AMPK is not well understood. Tissue culture medium alone contains high concentrations of glucose, to which 5–10% FBS is typically added, which results in excessive growth stimulation. This may explain why it is necessary to use higher doses than those used for patients with diabetes to observe the antitumor effects of metformin in cell culture systems [12]. There have been concerns regarding the determination of optimal therapeutic concentrations required to observe the cellular effects of metformin in preclinical models [40].

One of the potential mechanisms of metformin activity is inhibition of the mTOR pathway. LKB1 is the main activator of AMPK under metabolic stress conditions. The literature suggests that the anticancer activity of metformin probably results from insulin-dependent and insulin-independent mechanisms [35]. The primary mechanism underlying the reliance of metformin on insulin involves reduction in insulin levels, which leads to diminished insulin binding to the insulin receptor (IR), thereby impeding tumor growth. Independent of insulin, metformin triggers AMPK activation by suppressing the mitochondrial complex I, resulting in compromised mitochondrial function, reduced adenosine triphosphate (ATP) production, and elevated levels of adenosine monophosphate (AMP). Cancer cells lacking LKB1 protein expression do not respond to metformin in vitro, suggesting that metformin triggers various cellular responses that may vary depending on the cancer’s origin [41]. The level of LKB1 expression elicited by immunohistochemical staining was reported to be decreased in patients with GC and associated with invasion and metastasis [42]. However, mutation of LKB1 is relatively rare (3%) and almost mutually exclusive to HER2 amplification in patients with GC when queried in the cBioPortal database. Due to the high incidence of LKB1 mutations in NSCLC, one group tested the activity of metformin on NSCLC cells and found that the effect of metformin was strictly dependent on LKB1 mutation status [43]. However, we did not detect significant mutations in LKB1 in the 37 GC cell lines. Patients with GC, especially those with HER2+ status, are highly likely to have wild-type LKB1, which means that the LKB1/AMPK pathway is intact and the activity of AMPK activator metformin is not limited. A recent phase II trial of metformin in individuals with oral premalignant lesions provided evidence that metformin administration resulted in encouraging histological responses and mTOR pathway modulation [13]. The authors showed that metformin induced a significant decrease in cell proliferation in the squamous epithelium and also a significant decrease in mTOR activity, which was assessed by pS6 staining. These data suggest that the synergistic effect of metformin could be associated with inhibition of mTOR pathway in HER2+ cancer cells.

## 5. Conclusions

Our results show that metformin inhibits HER2+ GC cell proliferation and tumor growth, possibly by suppressing HER2 phosphorylation and inhibiting the mTOR pathway. Metformin is widely used for the treatment of type 2 diabetes mellitus with limited adverse effects. Therefore, metformin may be a safe and effective therapy for treatment of HER2+ GC, providing additional benefits at a low cost. In the era of targeted agents and immune checkpoint inhibitors, it is almost impossible to perform clinical trials using old drugs such as metformin. Currently, there are no clinical trials with metformin in patients with GC; however, there are phase III and IV clinical trials in other types of cancers (brain, breast, oral, and prostate cancers). Further studies are warranted in patients with GC, especially those with HER2 positive status. Although metformin has been used in clinical practice for over 60 years, new mechanisms relevant to cancer biology are still emerging. Well-curated, large-scale, real-world evidence could provide supporting data for combining Tmab and metformin for the management of patients with HER2+ GC.

## Figures and Tables

**Figure 1 cancers-15-04768-f001:**
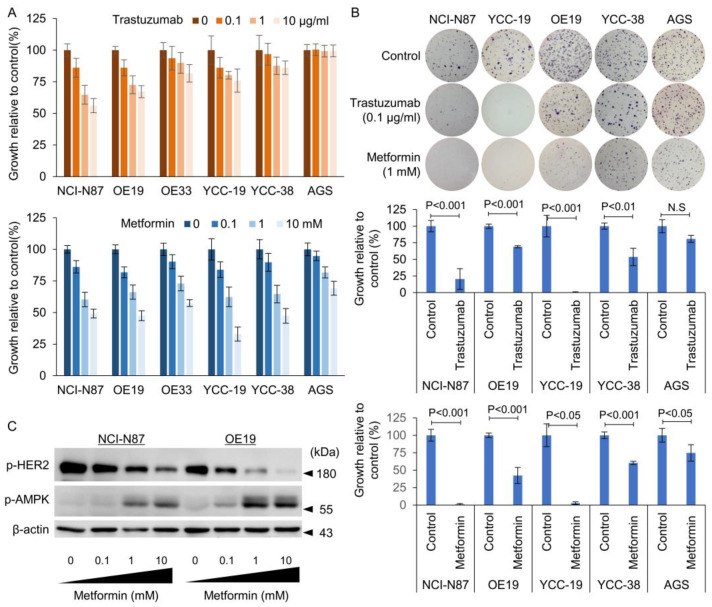
Dose-dependent effect of trastuzumab (Tmab) and metformin on HER2-positive and HER2-negative gastric cancer cells. (**A**) Tmab and metformin were added at specified concentrations to HER2-positive and HER2-negative gastric cancer cell lines and MTT assays were performed after 72 h (*n* = 3); (**B**) Tmab and metformin were added at a single concentration of 0.1 μg/mL and 1 mM for 3 weeks on soft agar colonies; (**C**) metformin induced phosphorylation of AMP kinase and dephosphorylation of HER2 in NCI-N87 and OE19 cells in a dose-dependent manner. The uncropped blots are shown in Appendix A.

**Figure 2 cancers-15-04768-f002:**
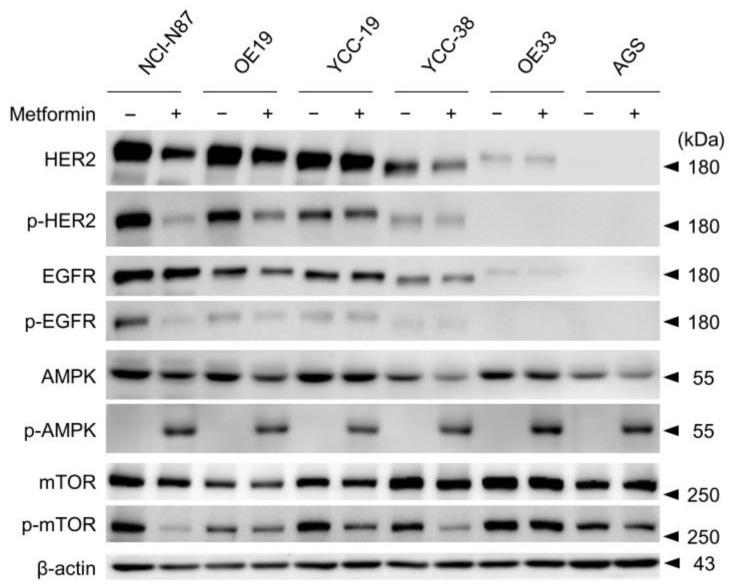
The cellular activity of metformin in gastric cancer cells with HER2-positive and HER2-negative cells, assessed using immunoblot analysis. Immunoblot analysis using antibodies specific to proteins indicated lysates of gastric cancer cells that were treated with metformin (10 mM) for 24 h before harvest. The combination of metformin resulted in decreased phosphorylation of HER2 and downstream targets, such as AKT or ERK, in Tmab-sensitive HER2+ cells. The uncropped blots are shown in Appendix A.

**Figure 3 cancers-15-04768-f003:**
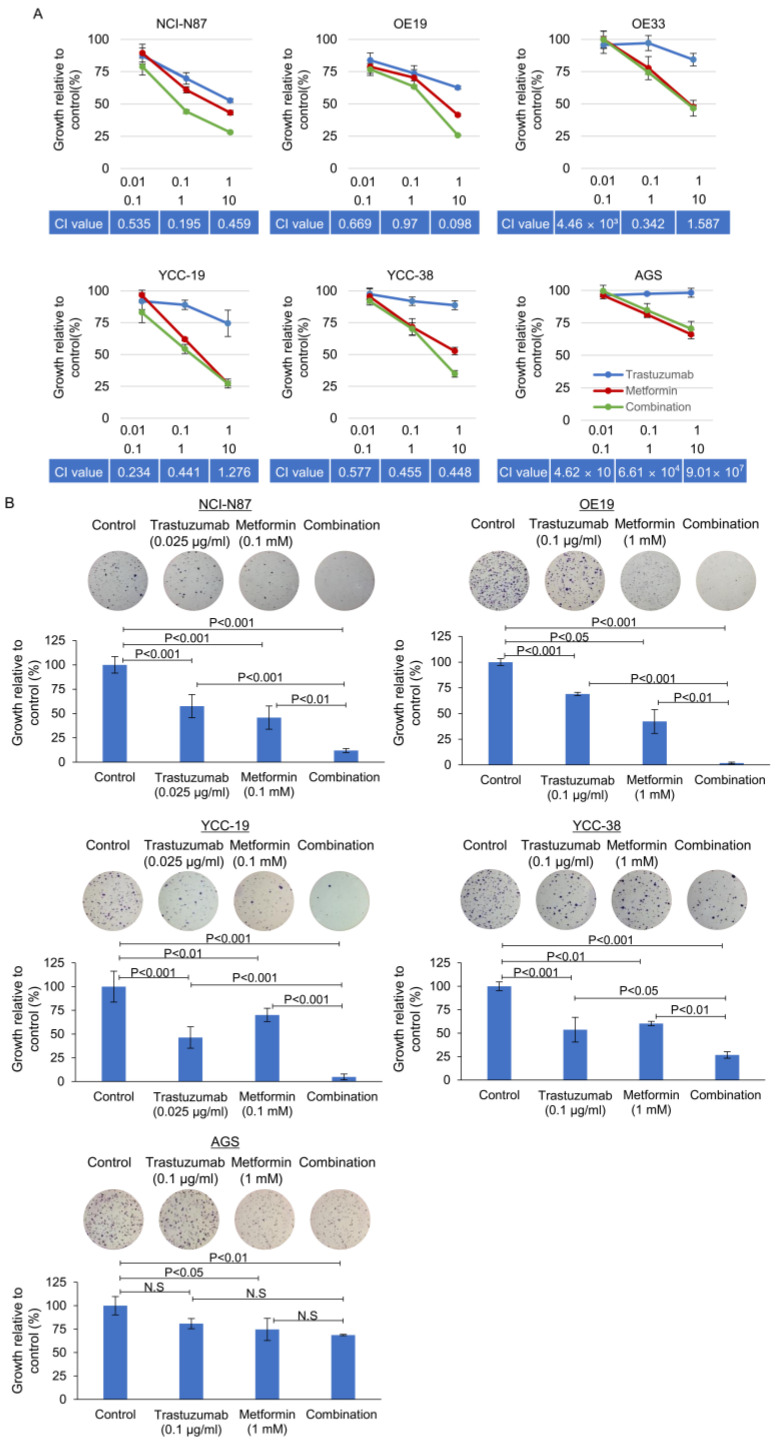
Growth inhibition, assessed by MTT assay (**A**) and soft agar assay (**B**), of HER2-positive and HER2-negative gastric cancer cells. Gastric cancer cells were treated with Tmab, metformin, or the combination as indicated concentrations. Combination indices are described at each curve (**A**). The effect of Tmab alone or in combination with metformin on anchorage-independent growth was determined by using soft-agar colony formation assay, as described in the Materials and Methods section. Tmab was administered at different concentrations (0.025 ug/mL or 0.1 ug/mL) based on Tmab sensitivity. MGI (mean growth inhibition) was calculated by dividing the colony number of the treated group by that of the untreated control group and are summarized in Table 1.

**Figure 4 cancers-15-04768-f004:**
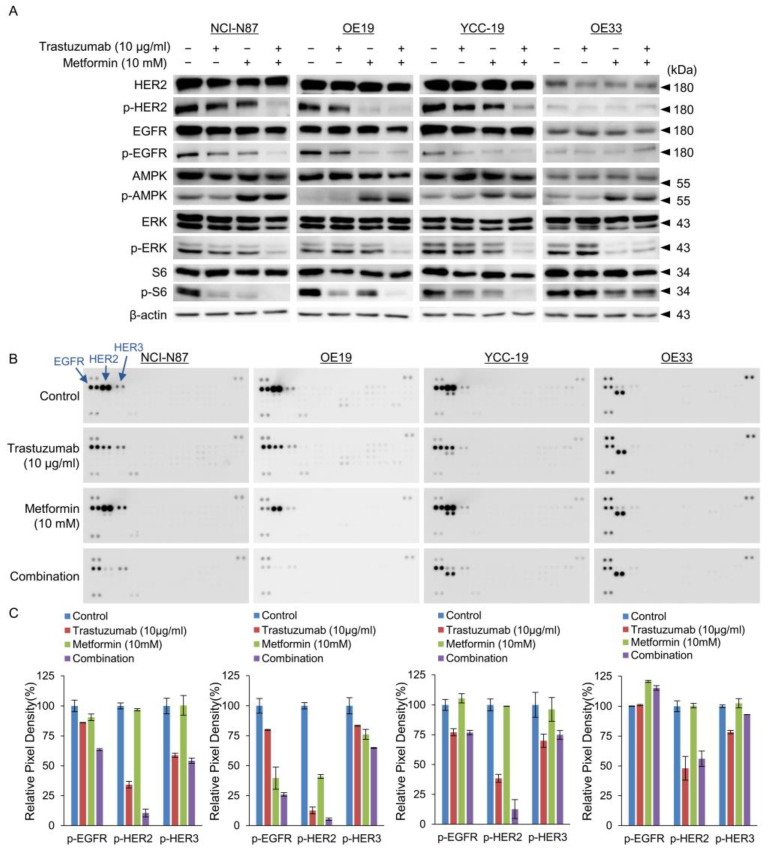
The cellular activity of trastuzumab and/or metformin in HER2 signal pathway, assessed using Immunoblot analysis (**A**), phospho-RTK antibody array (**B**), and relative pixel density from RTK arrays (**C**). The combination of Tmab and metformin resulted in decreased phosphorylation of HER2 and downstream targets, such as AKT or ERK, in Tmab-sensitive HER2+ cells. Phospho RTK arrays showed synergistic decrease in phosphorylation of EGFR, HER2 and HER3 with Tmab and metformin after 24 h exposure. The relative RTK activation in Tmab or metformin-treated cells were determined by dividing the density of each spot in the treated group by that of the corresponding spot in the vehicle-treated group, after normalizing to the average background signal. The uncropped blots are shown in Appendix A.

**Figure 5 cancers-15-04768-f005:**
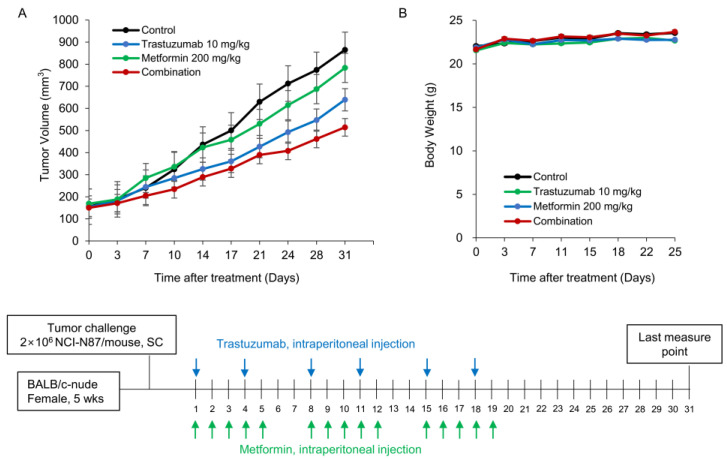
Effects of combination treatment of trastuzumab and metformin on HER2-positive gastric cancer NCI-N87 xenograft (**A**). The body weight of the mice was similar among the groups (**B**). The dosing schedule for Tmab and metformin is depicted at the bottom. From D24 to D31, the sizes of the xenografts among the four groups differed significantly (D24, *p* = 0.02; D28, *p* = 0.023; D31, *p* = 0.03, Kruskal–Wallis Test). The combination of Tmab and metformin showed significant tumor growth reduction in comparison to control (*p* = 0.021, ANOVA test) and metformin-treated group (*p* = 0.021, ANOVA test) from D24.

**Table 1 cancers-15-04768-t001:** The synergistic effect of Trastuzumab in combination with metformin on inhibition of anchorage-independent colony formation.

Cell Lines	Trastuzumab		Metformin		Combination			Index §
	MGI *	*p* †	MGI	*p* †	Expected	Observed	*p* †	
N87	0.575	<0.001	0.458	<0.001	0.264	0.120	<0.001	2.189
OE19	0.690	<0.001	0.423	<0.001	0.292	0.019	<0.001	15.621
YCC-19	0.465	<0.001	0.702	0.004	0.326	0.049	<0.001	6.666
YCC-38	0.536	<0.001	0.603	<0.001	0.324	0.267	<0.001	1.211
AGS	0.808	0.106	0.747	0.017	0.604	0.687	0.003	0.879

* MGI (mean growth inhibition) = colony number of treated group/colony number of untreated group. † *p*-value were calculated by Tukey’s HSD test compared to control group. § was calculated by dividing the expected colony number by the observed colony number. An index >1 indicates a synergistic effect, and an index <1 indicates less than additive effect.

## Data Availability

The data presented in this study are available upon request from the corresponding author.

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
