# Peer review of "Synergistic Effects of Metformin and Trastuzumab on HER2 Positive Gastroesophageal Adenocarcinoma Cells In Vitro and In Vivo"

_cancers, 2023, doi:10.3390/cancers15194768_

Round 1

Reviewer 1 Report

Thank you for the opportunity to review the manuscript titled "Synergistic effects of metformin and trastuzumab for HER2 positive gastroesophageal adenocarcinoma cells in vitro  and in vivo" (cancers-2606105). I find the subject of the article exciting; however, there are several areas where the text could be improved. I believe that with major revisions, this work has the potential to be accepted for publication. Please find my detailed comments below:

Title:

  1. The title is clear and informative.

Simple Summary:

  1. The simple summary provides a concise overview of the research, which is good for general understanding.

Abstract:

  1. Background: Specify the precise number of GC cases where HER2 amplification is observed. For example, "HER2 amplification occurs in approximately X% of GC cases."
  2. Methods: Mention briefly how metformin was administered to the cells in vitro. Was it tested as a single agent or in combination with trastuzumab? This will clarify the experimental setup.
  3. Results: Provide more specific quantitative results. For instance, how much of a decrease in the number of colonies was observed? Include numerical data to support the findings.
  4. Phospho-RTK arrays: Explain what RTK stands for and provide a brief description of the technology for clarity.
  5. Conclusion: The conclusion is clear but could be more specific. State the practical implications of this synergy between metformin and trastuzumab in HER2+ GC more explicitly. What are the potential clinical applications or future research directions?

Keywords:

  1. Keywords are appropriate and relevant.

Introduction:

  1. The introduction provides a good context for the study, but it can be made more concise.
  2. Avoid redundancy by not restating information mentioned in the abstract.
  3. When mentioning clinical trials like ToGA, provide the full names and references in a citation to facilitate further reading.
  4. Clarify what is meant by "medical unmet need for systemic treatment for patients with HER2+GC." Specify the challenges or gaps in current treatment options for these patients.
  5. Provide a more direct link between the altered signaling pathways like PI3K/AKT/mTOR and the rationale for using metformin. How does metformin potentially impact these pathways in GC?
  6. While you briefly mention the mechanisms of action of metformin, consider expanding slightly on how these mechanisms may be relevant to GC treatment.
  7. Include references to specific studies that support the anticarcinogenic activity of metformin, especially those related to HER2+ GC.
  8. State the research objectives or hypotheses more explicitly. What specific questions is this study aiming to answer?

Materials and Methods:

Cell Line and Reagents:

  1. Specify the total number of cell lines within the 37 mentioned that were HER2-positive. This information will help in understanding the selection process.
  2. It would be beneficial to provide some characteristics or key features of these selected cell lines, especially the gastric cancer cell lines. For instance, are they derived from different patient types, stages, or genetic backgrounds?
  3. Mention the passage numbers or any specific culture conditions that might be relevant to the experiments.

Droplet Digital PCR:

  1. Define abbreviations like "ddPCR" upon first use for clarity.
  2. The primer sequences are said to be in Supplementary Table S2, but there is no reference to this table. Ensure the table is included, or if not, provide the primer sequences directly.
  3. Clarify whether the HER2 copy number analysis was performed for all the selected cell lines or only for a subset. Also, explain the relevance of this analysis in the context of your study.

Cell Proliferation Assay:

  1. Mention the units for the concentrations of Tmab and metformin used. For example, "indicated concentrations of Tmab (in µg/ml) or metformin (in mM) were used."
  2. Describe the rationale for choosing the specific concentrations tested. Why were these concentrations considered relevant or physiologically significant?

Growth in Soft Agar:

  1. Specify the time period for colony formation assay incubation. For example, "The plates were incubated for 10–15 days."
  2. Clarify whether metformin and Tmab were added continuously during the incubation period or only at the beginning.
  3. Provide more detail on the concentration range of metformin and Tmab used in this assay.
  4. Define "Mean growth inhibition (MGI)" and "index" in a concise manner upon first use.

Western Blot Analysis:

  1. Mention the specific cell lysate preparation protocol briefly. For example, "Cells were lysed using RIPA buffer containing protease and phosphatase inhibitors."
  2. Provide more context regarding why these specific signaling pathway proteins were chosen for analysis. What is their relevance to the study?
  3. Clarify the units used for protein quantification (e.g., μg of protein loaded per lane).
  4. Mention how the protein expression was normalized (e.g., relative to actin) and whether any loading controls were used.

Proteome Profiling with Phospho-RTK Array:

  1. Describe the significance of analyzing receptor tyrosine kinases (RTKs) in the context of this study.
  2. Explain how the fold change was calculated in more detail.

Xenograft Mouse Model:

  1. Specify the gender of the mice used. For instance, "Five-week-old female athymic nude mice were obtained..."
  2. Include information on the source or supplier of the mice.
  3. Clarify the rationale behind the chosen dosage and administration schedule for Tmab and metformin. Why were these specific regimens selected?
  4. Provide information about the humane endpoints or criteria used for monitoring the health of the mice during the experiment.

Statistical Analysis:

  1. Mention if data normality tests were performed before selecting the appropriate statistical tests.
  2. Clarify which specific statistical tests were used for each type of analysis (e.g., ANOVA for multiple group comparisons).
  3. Define the "combination index (CI)" and briefly explain how it was calculated. It's important to ensure that readers unfamiliar with this term can understand its meaning.

Discussion:

Synergy between Metformin and Tmab:

  1. Specify the observed degree of synergy. Was this quantified, e.g., by a combination index (CI) value? Including quantitative data on synergy would provide more robust support for your findings.
  2. Mention the potential clinical implications of this synergy. How might it impact the treatment of HER2+ GC patients? Does it suggest a possible new treatment strategy?

Comparison with HER2+ Breast Cancer (BC):

  1. When discussing the differences in response to HER2-targeted agents between HER2+ GC and BC, consider briefly addressing possible reasons or hypotheses, even if they are beyond the scope of your study. This can enhance the comprehensiveness of your discussion.

Mechanisms of Metformin Activity:

  1. While you discuss the effect of metformin on HER2 phosphorylation and the mTOR pathway, consider expanding on the molecular mechanisms involved. For example, how does metformin modulate these pathways? Any insights into the signaling cascades would be valuable.
  2. You mention that metformin has insulin-dependent and insulin-independent mechanisms; elaborate on these mechanisms briefly.
  3. To improve clarity, consider a separate section that focuses on the potential mechanisms of metformin's activity in HER2+ GC cells.

Discussion of LKB1 and AMPK:

  1. Clarify the relevance of LKB1 and AMPK to your study. How does their status or activity impact the response to metformin and Tmab in HER2+ GC cells?
  2. Discuss the potential clinical implications of your findings related to LKB1 and AMPK. Do they suggest specific patient populations that might benefit more from this combination therapy?

Metformin Dosing in Cell Culture:

  1. Provide a brief explanation for the need to use higher metformin doses in cell culture compared to clinical use. Is this related to the glucose-rich environment in culture?

Conclusions:

  1. Emphasize the clinical relevance of your findings in the concluding remarks. What could this mean for the future of HER2+ GC treatment?
  2. Although you mention there are no clinical trials with metformin on GC patients currently, consider discussing any potential future directions or challenges for translating your research into clinical practice.

Moderate editing of English language required

Author Response

Please find the response to the reviewer regarding our manuscript entitled “Synergistic effects of metformin and trastuzumab for HER2 positive gastroesophageal adenocarcinoma cells in vitro and in vivo” for your reconsideration for publication in Cancers. We appreciate the thoughtful and comprehensive remarks of the reviewer and editors.  Please find below a point-by point response to the comments. We hope our work is suitable for publication in Cancers and look forward to hearing from you after a decision has been made.

Sincerely,

Jin-Soo Kim

Department of Internal Medicine

Seoul Metropolitan Government Seoul National University Boramae Medical Center

Seoul National University College of Medicine

20 Boramae-ro 5-gil, Dongjak-gu, Seoul 07061, South Korea

Tel: 82-2-870-3202

Fax: 82-2-831-2826

Reviewer 1

Thank you for the opportunity to review the manuscript titled "Synergistic effects of metformin and trastuzumab for HER2 positive gastroesophageal adenocarcinoma cells in vitro  and in vivo" (cancers-2606105). I find the subject of the article exciting; however, there are several areas where the text could be improved. I believe that with major revisions, this work has the potential to be accepted for publication. Please find my detailed comments below:

Title:

The title is clear and informative.

Simple Summary:

The simple summary provides a concise overview of the research, which is good for general understanding.

Abstract:

Background: Specify the precise number of GC cases where HER2 amplification is observed. For example, "HER2 amplification occurs in approximately X% of GC cases."

Methods: Mention briefly how metformin was administered to the cells in vitro. Was it tested as a single agent or in combination with trastuzumab? This will clarify the experimental setup.

Results: Provide more specific quantitative results. For instance, how much of a decrease in the number of colonies was observed? Include numerical data to support the findings.

Phospho-RTK arrays: Explain what RTK stands for and provide a brief description of the technology for clarity.

Conclusion: The conclusion is clear but could be more specific. State the practical implications of this synergy between metformin and trastuzumab in HER2+ GC more explicitly. What are the potential clinical applications or future research directions?

Keywords:

Keywords are appropriate and relevant.

Response>

Abstract has been revised as the reviewer requested. We had 6 cell lines and 4 different conditions (control, Tmab, metformin and combination), so we described the decrease in the combo treatment briefly.

Introduction:

The introduction provides a good context for the study, but it can be made more concise.

Avoid redundancy by not restating information mentioned in the abstract.

Response>

We have revised the introduction as the reviewer suggested.

When mentioning clinical trials like ToGA, provide the full names and references in a citation to facilitate further reading.

Response>

The naming of clinical trials is not necessarily an acronym of the title of clinical trials. Sometimes, the authors did not clarify the reason of the naming. We removed the names of the clinical trial other than ToGA.

Clarify what is meant by "medical unmet need for systemic treatment for patients with HER2+GC." Specify the challenges or gaps in current treatment options for these patients.

Response>

We have revised the introduction as the reviewer suggested. Medical unmet needs for HER2+ GC originated from the lack of efficacy of other HER2 targeted agents, which have been successful for HER2+ BC.

Provide a more direct link between the altered signaling pathways like PI3K/AKT/mTOR and the rationale for using metformin. How does metformin potentially impact these pathways in GC?

While you briefly mention the mechanisms of action of metformin, consider expanding slightly on how these mechanisms may be relevant to GC treatment.

Include references to specific studies that support the anticarcinogenic activity of metformin, especially those related to HER2+ GC.

State the research objectives or hypotheses more explicitly. What specific questions is this study aiming to answer?

Response>

We started this study because metformin for HER2 + GC has not been studied well. We have searched the literature again and add a few references to support the use of metformin in GC as the reviewer suggested. Exact mechanisms of metformin are not well understood and further investigation like the our work is needed.

Materials and Methods:

Cell Line and Reagents:

Specify the total number of cell lines within the 37 mentioned that were HER2-positive. This information will help in understanding the selection process.

It would be beneficial to provide some characteristics or key features of these selected cell lines, especially the gastric cancer cell lines. For instance, are they derived from different patient types, stages, or genetic backgrounds?

Mention the passage numbers or any specific culture conditions that might be relevant to the experiments.

Response>

In the original submission, from line 205-206, we already described that 10 GC cells showed HER2 positivity. In the supplementary table S1, we showed the source of the tested GC cells. These cells were purchased from public cell banks or given by prof. Rha, so we could only retrieve the information from the vendors. For cell cultures, we recorded passage numbers for each cell lines and kept passage numbers low.

Droplet Digital PCR:

Define abbreviations like "ddPCR" upon first use for clarity.

The primer sequences are said to be in Supplementary Table S2, but there is no reference to this table. Ensure the table is included, or if not, provide the primer sequences directly.

Clarify whether the HER2 copy number analysis was performed for all the selected cell lines or only for a subset. Also, explain the relevance of this analysis in the context of your study.

Response>

The droplet digital PCR (ddPCR) was first described in line 110 of the original manuscript, and abbreviation of “ddPCR” was used after this. We have submitted all supplementary files in compressed file. You can find the Supplementary Table S2 in the zip file. All of the 37 cell lines was tested for HER2 copy numbers and the results were shown in the Supplementary Figure S1 in the zip file. We are afraid that the reviewer’s questions are already described in the original submission.

Cell Proliferation Assay:

Mention the units for the concentrations of Tmab and metformin used. For example, "indicated concentrations of Tmab (in µg/ml) or metformin (in mM) were used."

Describe the rationale for choosing the specific concentrations tested. Why were these concentrations considered relevant or physiologically significant?

Response>

We have added the units for each agent in the revision. As we described in line 216-218 and Supplementary Table S3 in the original submission, most HER2+ GC cells did not reach IC50 even at 10 μg/ml with trastuzumab. As described in Figure 1A, we tested several concentrations of Tmab and metformin for these cells. Based on these observations, we selected the specific concentrations for the subsequent experiments.

Growth in Soft Agar:

Specify the time period for colony formation assay incubation. For example, "The plates were incubated for 10–15 days."

Clarify whether metformin and Tmab were added continuously during the incubation period or only at the beginning.

Provide more detail on the concentration range of metformin and Tmab used in this assay.

Define "Mean growth inhibition (MGI)" and "index" in a concise manner upon first use.

Response>

Time period is already described in line 136-137 in the original submission. The cells were treated with each agent twice a week during the media refreshment. Concentration range of metformin and Tmab was described in the Results in the original submission where appropriate. The definition of MGI and how to calculate the MGI are already described in line 140-145 in the original submission.

Western Blot Analysis:

Mention the specific cell lysate preparation protocol briefly. For example, "Cells were lysed using RIPA buffer containing protease and phosphatase inhibitors."

Provide more context regarding why these specific signaling pathway proteins were chosen for analysis. What is their relevance to the study?

Clarify the units used for protein quantification (e.g., μg of protein loaded per lane).

Mention how the protein expression was normalized (e.g., relative to actin) and whether any loading controls were used.

Response>

We mentioned that “The proteins were harvested as previously described” and cited reference for this. We thought that we could reduce the redundancy or overlap with other article by citing the reference from the same author.  We described the preparation in brief as the reviewer recommended. Also, we described that “Western blots were performed to compare the expression of relevant signaling pathway, such as HER2 signaling pathway.” The chosen proteins are well known down-stream for HER2 signaling pathway.

Proteome Profiling with Phospho-RTK Array:

Describe the significance of analyzing receptor tyrosine kinases (RTKs) in the context of this study.

Explain how the fold change was calculated in more detail.

Response>

The purpose of phosphor-RTK array is to profile the proteome of growth factor pathway in brief. The significance of this analysis is described in the Results section. How to calculate the fold change is described in line 170-174 in the original submission.

Xenograft Mouse Model:

Specify the gender of the mice used. For instance, "Five-week-old female athymic nude mice were obtained..."

Include information on the source or supplier of the mice.

Clarify the rationale behind the chosen dosage and administration schedule for Tmab and metformin. Why were these specific regimens selected?

Provide information about the humane endpoints or criteria used for monitoring the health of the mice during the experiment.

Response>

We added “female” in the revision. The source of the mice is described in line 178 in the original submission. Since the animal experiments was performed after the approval by IACUC of Seoul National University, the animals were monitored and sacrificed accordingly.

Statistical Analysis:

Mention if data normality tests were performed before selecting the appropriate statistical tests.

Clarify which specific statistical tests were used for each type of analysis (e.g., ANOVA for multiple group comparisons).

Define the "combination index (CI)" and briefly explain how it was calculated. It's important to ensure that readers unfamiliar with this term can understand its meaning.

Response>

We are afraid that we can keep the current description of statistical analysis despite of the reviewer’s comments. Most experiments were repeated 3 to 5 times, so we had to use non-parametric test such as Kruskal-Wallis test or ANOVA test. Furthermore, CI is well known method to evaluate synergism since 1980’s and the software is commercially available. We added the original reference for CI method.

Discussion:

Synergy between Metformin and Tmab:

Specify the observed degree of synergy. Was this quantified, e.g., by a combination index (CI) value? Including quantitative data on synergy would provide more robust support for your findings.

Mention the potential clinical implications of this synergy. How might it impact the treatment of HER2+ GC patients? Does it suggest a possible new treatment strategy?

Response>

Degrees of synergy are different among the assays and cannot be summarized in few sentences. We added the degree of tumor reduction from the xenograft experiment. CI is mostly used to describe the synergism on MTT assays, but this is not appropriate to measure the synergism on xenograft experiments or Western blots. The potential clinical implications and future suggestions are discussed in the last section of Discussion.

Comparison with HER2+ Breast Cancer (BC):

When discussing the differences in response to HER2-targeted agents between HER2+ GC and BC, consider briefly addressing possible reasons or hypotheses, even if they are beyond the scope of your study. This can enhance the comprehensiveness of your discussion.

Response>

As the reviewer suggested, we agree that this is very interesting research topic. But the present study did not involve HER2+ breast cancers. The differential sensitivity to anti-HER2 agents between HER2+ BC and GC are partly discussed in a few review articles, and we included the references in the revised manuscript.

Mechanisms of Metformin Activity:

While you discuss the effect of metformin on HER2 phosphorylation and the mTOR pathway, consider expanding on the molecular mechanisms involved. For example, how does metformin modulate these pathways? Any insights into the signaling cascades would be valuable.

You mention that metformin has insulin-dependent and insulin-independent mechanisms; elaborate on these mechanisms briefly.

To improve clarity, consider a separate section that focuses on the potential mechanisms of metformin's activity in HER2+ GC cells.

Response>

The exact molecular mechanism of the modulation of HER2 activity by AMPK activation is not well understood. As discussed in the original submission, metformin can reduce the activity of growth factor pathway by activating AMPK. Several articles suggested that activated AMPK could modulate growth factor signaling in breast cancer cell line models, but there is scarce data for HER2+ GC models. Insulin-dependent and independent mechanism of metformin are briefly discussed as the reviewer suggested. We were not able to elaborate on the potential mechanisms of metformin activity specifically for HER2+ GC, as there is a limited amount of data accessible for for this aspect.

Discussion of LKB1 and AMPK:

Clarify the relevance of LKB1 and AMPK to your study. How does their status or activity impact the response to metformin and Tmab in HER2+ GC cells?

Discuss the potential clinical implications of your findings related to LKB1 and AMPK. Do they suggest specific patient populations that might benefit more from this combination therapy?

Response>

As discussed in the original submission, LKB1 is the main activator of AMPK. But mutation of LKB1 was relatively rare (3%) and almost mutually exclusive with HER2 amplification in GC patients when we queried in cBioPortal database. The potential interaction between LKB1 and HER2 is mostly speculative now and discussed in the manuscript. Since the LKB1 mutation is very rare in GC and does not seem to co-occur with HER2 amplification, LKB1 is not important biomarker for HER2+ GC. We described this in the manuscript as “GC patients, especially for HER2+ cases, ……., the activity of AMPK activator metformin will not be limited.”

Metformin Dosing in Cell Culture:

Provide a brief explanation for the need to use higher metformin doses in cell culture compared to clinical use. Is this related to the glucose-rich environment in culture?

Response>

In line 347-350 of the original submission, we cited a reference for the higher dose of metformin in cell culture system. As the reviewer suspected, there have been concerns about the concentration of metformin in the pre-clinical models, and we added a sentence based on a recent review article.

Conclusions:

Emphasize the clinical relevance of your findings in the concluding remarks. What could this mean for the future of HER2+ GC treatment?

Although you mention there are no clinical trials with metformin on GC patients currently, consider discussing any potential future directions or challenges for translating your research into clinical practice.

Response>

As you suggested, we elaborate more on the potential directions of future studies.

Reviewer 2 Report

Please take a look at the file I've attached for details.

Journal: Cancers

Title: Synergistic effects of metformin and trastuzumab for HER2 positive gastroesophageal adenocarcinoma cells in vitro and in vivo

The manuscript evaluates the role of metformin in combination with the clinical drug trastuzumab in HER2+ gastric cancer cell lines. The cells treated with various concentrations of metformin, trastuzumab, and their combination were assessed for cell proliferation using MTT assay, colony formation on agar, Western blot for AMP kinase, and mTOR pathway. All these studies revealed synergistic effects on cells with metformin in combination with trastuzumab. In vivo xenograft study on GC mouse model also showed a synergistic effect when treated with metformin in combination with trastuzumab, giving credence to the hypothesis that metformin could be used to treat HER2+ GC patients in combination with standard therapy. The manuscript is well-written, and the overall flow is good. However, minor revisions must be made to improve the overall quality and conclusiveness of the results. Please see the comments below for revisions.

Minor revisions:

1. The authors mentioned which cells were evaluated in the abstract methods (line 21). Please include the method by which the cells were evaluated.

2. In the materials and methods (line 114), please provide the source of primers and probes.

3. For cell proliferation assay (lines 128-132), please provide a detailed method or indicate that the method was followed according to the manufacturer’s protocol. Also, include which plate reader was used to record the absorbance.

4. 0.001% crystal violet was used to stain the colonies. Indicate which solvent crystal violet is dissolved in (water?)

5. For Western blot assays, include which membrane type was used (nitrocellulose or PVDF). Also, include how the chemiluminescence was recorded (Bio-Rad, licor, or other source).

6. In the xenograft model, 5-week-old athymic nude mice were used. Please include whether male, female, or a combination of mice were used. The rationale for using the NCI-N87 cell line for this study must be included.

7. To support the results of supplementary table S3, please include the representative IC50 graphs for trastuzumab and metformin.

8. The manuscript has many grammatical errors that should be corrected. Below are some of the examples. The manuscript should undergo thorough proofreading.

Line 29: “Phospho-RTK arrays showed that the synergistic decrease of phosphorylation of EGFR, HER2 and HER3” should be “Phospho-RTK arrays showed that there is a synergistic decrease of phosphorylation of EGFR, HER2 and HER3”

Line 87: “diabetes patients” should be changed to “diabetic patients”

Line 134: 2.5 × 103 to 7.5 × 103. Indicate the power appropriately as “2.5 × 10^3 to 7.5 × 10^3”

Line 180-182: “The mice were injected subcutaneously into the right flank with viable NCI-N87 at 2×106 cells/mouse” should be “The cells were injected subcutaneously into the right flank with viable NCI-N87 at 2×106 cells/mouse”

Please take a look at the attached review for details.

Author Response

Please find the response to the reviewer regarding our manuscript entitled “Synergistic effects of metformin and trastuzumab for HER2 positive gastroesophageal adenocarcinoma cells in vitro and in vivo” for your reconsideration for publication in Cancers. We appreciate the thoughtful and comprehensive remarks of the reviewer and editors.  Please find below a point-by point response to the comments. We hope our work is suitable for publication in Cancers and look forward to hearing from you after a decision has been made.

Sincerely,

Jin-Soo Kim

Department of Internal Medicine

Seoul Metropolitan Government Seoul National University Boramae Medical Center

Seoul National University College of Medicine

20 Boramae-ro 5-gil, Dongjak-gu, Seoul 07061, South Korea

Tel: 82-2-870-3202

Fax: 82-2-831-2826

Reviewer 2

The manuscript evaluates the role of metformin in combination with the clinical drug trastuzumab in HER2+ gastric cancer cell lines. The cells treated with various concentrations of metformin, trastuzumab, and their combination were assessed for cell proliferation using MTT assay, colony formation on agar, Western blot for AMP kinase, and mTOR pathway. All these studies revealed synergistic effects on cells with metformin in combination with trastuzumab. In vivo xenograft study on GC mouse model also showed a synergistic effect when treated with metformin in combination with trastuzumab, giving credence to the hypothesis that metformin could be used to treat HER2+ GC patients in combination with standard therapy. The manuscript is well-written, and the overall flow is good. However, minor revisions must be made to improve the overall quality and conclusiveness of the results. Please see the comments below for revisions.

Minor revisions:

  1. The authors mentioned which cells were evaluated in the abstract methods (line 21). Please include the method by which the cells were evaluated.

Response>

We have updated the abstract as the reviewer suggested.

  1. In the materials and methods (line 114), please provide the source of primers and probes.

Response>

We clarified that the reference genes primer-probe set for ddPCR were designed by LOGONE Bio-Convergence Research Foundation (Seoul, South Korea).

  1. For cell proliferation assay (lines 128-132), please provide a detailed method or indicate that the method was followed according to the manufacturer’s protocol. Also, include which plate reader was used to record the absorbance.

Response>

We have added “according to the manufacturer’s protocol”. Plate reader is specified in the revision.

  1. 0.001% crystal violet was used to stain the colonies. Indicate which solvent crystal violet is dissolved in (water?)

Response>

Crystal violet was purchased and diluted in distilled water and we added the information in the revision.

  1. For Western blot assays, include which membrane type was used (nitrocellulose or PVDF). Also, include how the chemiluminescence was recorded (Bio-Rad, licor, or other source).

Response>

Type of membrane and the chemiluminescence machine are specified in the revision.

  1. In the xenograft model, 5-week-old athymic nude mice were used. Please include whether male, female, or a combination of mice were used. The rationale for using the NCI-N87 cell line for this study must be included.

Response>

We clarified that we used “female” mice for the xenograft experiments. NCI-N87 cells are the most used HER2+ GC cells for xenografts. We added the description in the revised manuscript.

  1. To support the results of supplementary table S3, please include the representative IC50 graphs for trastuzumab and metformin.

Response>

We have replaced the Supplementary Table S3 with Supplementary Figure S2 to show the IC50 graphs in the revision.

  1. The manuscript has many grammatical errors that should be corrected. Below are some of the examples. The manuscript should undergo thorough proofreading.

Line 29: “Phospho-RTK arrays showed that the synergistic decrease of phosphorylation of EGFR, HER2 and HER3” should be “Phospho-RTK arrays showed that there is a synergistic decrease of phosphorylation of EGFR, HER2 and HER3”

Line 87: “diabetes patients” should be changed to “diabetic patients”

Line 134: 2.5 × 103 to 7.5 × 103. Indicate the power appropriately as “2.5 × 10^3 to 7.5 × 10^3”

Line 180-182: “The mice were injected subcutaneously into the right flank with viable NCI-N87 at 2×106 cells/mouse” should be “The cells were injected subcutaneously into the right flank with viable NCI-N87 at 2×106 cells/mouse”

Response>

We have corrected the grammatical errors and re-write several sentences as the reviewer recommended.

Round 2

Reviewer 1 Report

Accepted for publication.